# Sericin Enhances Cryopreserved Sperm Quality in Chengde Hornless Black Goats by Increasing Glutamine Metabolism

**DOI:** 10.3390/ani15192830

**Published:** 2025-09-28

**Authors:** Yang Yu, Wei Xia, Wentao Zhang, Chenyu Tao, Xiaofeng Tian, Mengqi Li, Xiaosheng Zhang, Jinlong Zhang, Shunran Zhao, Yatian Qi, Tianmiao Qin, Junjie Li

**Affiliations:** 1College of Animal Science and Technology, Hebei Agricultural University, Baoding 071000, China; yyang980828@163.com (Y.Y.);; 2Hebei Technology Innovation Center of Cattle and Sheep Embryo, Baoding 071000, China; 3Tianjin Institute of Animal Sciences and Veterinary Medicine, Tianjin 300380, China

**Keywords:** Chengde hornless goat, sericin, sperm cryopreservation, antioxidant, sperm motility

## Abstract

**Simple Summary:**

The cryopreservation of Chengde polled goat semen is vital for genetic conservation and artificial insemination, but freezing-induced oxidative stress severely damages sperm motility. This study demonstrates that adding 0.6% (*w*/*v*) sericin to cryoprotectants significantly improves post-thaw sperm viability, while reducing sperm abnormalities and intracellular reactive oxygen species (ROS). Integrated proteomic and metabolomic analyses identified 162 differentially expressed proteins and 109 metabolites in sericin-treated sperm. Key findings suggest the involvement of glutamine in alanine/aspartate/glutamate metabolism pathways, alongside upregulated antioxidant proteins (LRP8, GSTM3, SIRT2). These results indicate that sericin may enhance cryotolerance by mitigating oxidative damage, sustaining energy metabolism, and improving structural integrity. The study offers insights into potential molecular mechanisms for optimizing semen cryopreservation protocols in goats.

**Abstract:**

The cryopreservation of Chengde polled goat semen plays a critical role in conserving genetic resources, enhancing the utilization efficiency of superior breeding bucks, and advancing artificial insemination techniques. However, spermatozoa are vulnerable to oxidative stress during the freezing process, which can significantly compromise sperm motility. In this study, pooled ejaculates from multiple bucks were divided into six groups, including a control group cryopreserved with conventional extender and five treatment groups supplemented with sericin at concentrations of 0.2%, 0.4%, 0.6%, 0.8%, and 1.0% (*w*/*v*). The results demonstrated that supplementation of the semen cryoprotectant with 0.6% sericin significantly improved post-thaw sperm viability to 65.25% in Chengde hornless goats, while concurrently reducing both the sperm abnormality rate (*p* < 0.05) and intracellular ROS levels (*p* < 0.05). Integrated TMT proteomics and LC/MS metabolomics further compared the 0.6% sericin group with the frozen control group and identified 162 differentially expressed proteins and 109 differential metabolites between the sericin supplementation and frozen control groups. Functional analysis revealed the significant enrichment of differential metabolites, such as glutamine, in the alanine, aspartate, and glutamate metabolism pathway, concomitant with the marked upregulation of antioxidant proteins including LRP8, GSTM3, and SIRT2. Thus, 0.6% sericin enhances cryotolerance primarily by improving sperm viability, reducing oxidative damage, and sustaining energy metabolism. These findings indicate that sericin enhances cryotolerance by reducing oxidative damage and supporting metabolic function, providing preliminary molecular insights for improving goat semen cryopreservation.

## 1. Introduction

The Chengde Hornless Black Goat, an indigenous Chinese breed, is distinguished by its robust adaptability and exceptional resilience. This breed exhibits rapid growth, substantial size, superior conformation, and high meat production performance, making it a valuable genetic resource for the selective breeding and improvement of other regional goat breeds [1]. However, factors such as prevalent disease pressures, widespread grazing restrictions, and low natural mating efficiency have collectively driven this breed towards an endangered status [2]. Furthermore, critical technical bottlenecks hinder its conservation and utilization. The cryopreservation of semen is compromised by a significant decline in post-thaw sperm quality and low conception rates following artificial insemination, as reported in studies on this breed [2]. These reproductive challenges significantly limit the breed’s broader implementation.

Semen cryopreservation is a vital technique for preserving animal genetic resources [3]. However, goat spermatozoa exhibit heightened sensitivity to cold shock compared to species such as humans, rabbits, dogs and cats, primarily due to their lower intra-membrane cholesterol–phospholipid ratios [4]. This sensitivity is further compounded by the higher abundance of polyunsaturated fatty acids (PUFAs) within the plasma membrane of caprine sperm [5]. Membrane rupture induced by these structural vulnerabilities impairs sperm viability and fertilizing capacity, consequently compromising reproductive efficiency [6]. Additionally, cryopreservation exposes sperm to severe oxidative damage and metabolic stress, particularly disrupting energy metabolism and osmotic regulation pathways essential for sperm function [7]. Therefore, incorporating antioxidant supplements into cryodiluents is crucial to mitigate reactive oxygen species (ROS) generation and associated oxidative stress injuries, thereby preserving post-thaw sperm quality and fertility potential. Sericin, a water-soluble glycoprotein extracted from silkworm (*Bombyx mori*) cocoons, has gained increasing attention as a cryoprotective agent due to its antioxidant properties and unique physicochemical characteristics [8]. Its high molecular weight and hydrophilic nature significantly increase the viscosity of cryoprotective solutions, potentially forming a protective film around sperm cells to mitigate ice crystal-induced physical damage [9]. Furthermore, sericin contributes to regulating the osmolality of the extender, thereby supporting osmotic balance during freezing and thawing processes [10,11]. The protein also exhibits mild pH-buffering capacity, which aids in maintaining extracellular stability [7]. These physicochemical properties complement its known antioxidant functions and collectively help sustain membrane integrity and reduce cryo-injury, making sericin a multi-functional additive for sperm cryopreservation [12,13].

Sericin has been successfully implemented in cryopreserving diverse mammalian cell types [14,15]. Several paper demonstrated that adding 0.25% (*w*/*v*) sericin to bovine semen cryoprotectants significantly improved post-thaw sperm quality by attenuating lipid peroxidation [16,17]. Ratchamak [18] and Chaturvedi [19] subsequently reported that 5 mg/mL sericin reduced oxidative damage in thawed bull semen, enhancing viability and capacitation status. Notably, beyond improving sperm quality parameters, sericin supplementation has been shown to enhance functional outcomes; a study on mouse sperm demonstrated that sericin not only improved post-thaw quality but also significantly increased fertilization ability and subsequent embryonic development potential [20].

While these studies consistently highlight the functional benefits of sericin, a comprehensive understanding of its mechanisms requires deeper molecular investigation. Although previous studies, such as that by Reddy et al. [21], have revealed alterations in gene expression after sericin supplementation, the corresponding proteomic and metabolic responses have not been comprehensively investigated. Building on evidence of sericin’s bioactive properties and the critical importance of metabolic regulation during cryopreservation, we specifically hypothesize that sericin enhances cryotolerance in Chengde hornless goat sperm by targeted modulation of key metabolic pathways, particularly glutamine and taurine metabolism, which are essential for maintaining cellular energy homeostasis, osmotic balance, and antioxidant defense during freeze–thaw stress [22]. Here, we employed TMT-based proteomics and LC/MS-based metabolomics to analyze cryopreserved semen from Chengde hornless goats supplemented with sericin. This study elucidates sericin’s cryoprotective mechanisms, providing a theoretical foundation for optimizing cryoprotectant formulations in this breed.

## 2. Materials and Methods

### 2.1. Animals

Six healthy bucks (aged 2–3 years) with comparable physique and proven reproductive fitness were selected. All animals were maintained under standardized breeding conditions, including a well-ventilated barn with natural lighting, seasonal temperature regulation, balanced diet, and regular health management. Semen was collected twice weekly via an artificial vagina during the breeding season (August to September 2023), following consistent protocols involving a preheated artificial vagina (45–50 °C), fixed morning collection times, and gentle handling to minimize stress. All animals were sourced from Chengde Fengning Manchu Autonomous Region Yuanxi Husbandry Co., Ltd. (Chengde, China). Experimental procedures strictly adhered to the Guide for the Care and Use of Agricultural Animals in Research and Teaching.

### 2.2. Semen Extender Supplemented with Sericin

The experimental design involved supplementing the commercial semen extender OptiXcell^®^ (026218, IMV, L’Aigle, France) with sericin (60650-88-6, Macklin, Shanghai, China). Freshly collected semen was pooled and randomly allocated to six groups: (1) a control group without sericin supplementation, and (2–6) experimental groups with sericin added to OptiXcell^®^ (026218, IMV, L’Aigle, France) at 0.2%, 0.4%, 0.6%, 0.8%, and 1.0% (*w*/*v*). During the cryopreservation procedure, the extended semen samples were prepared by mixing the fresh semen with the respective extender in a ratio of 10:1 (semen: extender, *v*/*v*). To assess the potential impact of sericin supplementation on the physicochemical properties of the highly concentrated extender, we measured the pH and osmotic pressure of the prepared extender solutions (before the addition of semen). The pH remained consistent at 6.4 across all groups. Although the osmolality values showed a numerical increasing trend with higher sericin concentration, the differences were not statistically significant (*p* ≥ 0.05, Table 1).

### 2.3. Semen Collection, Quality Assessment, and Isothermal Dilution

Semen was collected from six bucks via artificial vagina twice weekly over a six-week period during the breeding season (August to September 2023), yielding a total of 72 ejaculates. Immediately after collection, fresh ejaculates were evaluated based on the following quality criteria: creamy white coloration, ejaculate volume ≥ 0.7 mL, sperm motility ≥ 75%, and sperm density ≥ 2.5 × 10^9^/mL. Only ejaculates meeting all criteria were used in subsequent procedures. To minimize individual variability, qualified samples were pooled, thoroughly mixed, and divided into six equal aliquots. Prior to dilution, the extender was pre-warmed to 38 °C in a water bath for 10 min. The pooled semen was then subjected to isothermal dilution at a 1:10 (*v*/*v*) ratio with the pre-warmed extender, yielding a final sperm concentration of 8 × 10^7^ cells/mL.

### 2.4. Freezing and Thawing of Semen

Diluted semen was loaded into 0.25 mL plastic straws, sealed with polyvinyl alcohol powder, labeled, and gauze-wrapped. Straws were equilibrated at 4 °C for 2 h, then vapor-cooled for 10 min on a metal rack positioned 4 cm above liquid nitrogen (LN_2_), followed by immersion in LN_2_ for storage. For thawing, intact and properly sealed straws were removed from LN_2_ and immersed in 38 °C water for 30 s. After drying with absorbent paper, semen was transferred to centrifuge tubes and incubated at 38 °C for 5 min prior to analysis.

### 2.5. Sperm Motility Assessment

Sperm kinematics were evaluated using a computer-assisted sperm analysis (CASA) system (Hamilton-Thorne, Beverly, MA, USA; IVOS 12.3). The system was configured to capture between 1 and 50 image sequences per sample, with each sequence consisting of 30 consecutive frames acquired at 60 frames per second. The total analysis time for each image set was 10 s or less. Equipped with spatial resolution that allowed detection of particle diameters of 2 μm or larger, the system enabled precise tracking of sperm heads based on their inherent contrast and movement.

To ensure reliability and account for biological variability, the entire evaluation process was performed with five independent replicates for each experimental group using pooled sperm samples. The kinematic profiles of spermatozoa were observed at 400× magnification under a phase-contrast microscope. The system accurately measured sperm velocities within a range of 0 to 150 μm/s, encompassing the full spectrum of motile spermatozoa kinematic characteristics. All analyses were performed using a four-chamber 10 μm depth sperm counting slide (Nanning Mailang Technology Co., Ltd., Nanning, China), with samples maintained at 37 °C on a heated stage.

The recorded parameters included total motility (TM, %), straight-line velocity (VSL, μm/s), curvilinear velocity (VCL, μm/s), average path velocity (VAP, μm/s), and amplitude of lateral head displacement (ALH, μm).

### 2.6. Sperm Abnormality Analysis

Sperm deformity rates were quantified using Coomassie Brilliant Blue staining (P0003M, Beyotime, Shanghai, China). Post-thaw spermatozoa from each experimental group were fixed in 10 mL of 2.5% glutaraldehyde/PBS at room temperature for 10 min. After centrifugation (2000× *g*, 5 min), supernatants were discarded and pellets were resuspended in PBS. Following a second centrifugation (1500× *g*, 3 min), pellets were resuspended in 2 mL PBS to achieve a final concentration of approximately 1 × 10^7^ sperm/mL. A 10 μL aliquot of the washed sperm suspension was smeared onto pre-cleaned slides and air-dried. Slides were stained with Coomassie Brilliant Blue solution for 20 min, rinsed under running water for 30 s, and air-dried in darkness. To ensure reliability and account for biological variability, the entire staining and evaluation process was performed with five independent replicates for each experimental group, using pooled sperm samples. Duplicate slides per sample were examined under bright-field microscopy at 200× magnification, with ≥200 sperm evaluated per slide by investigators blinded to treatment groups. Deformity rate was calculated as: (Number of abnormal sperm/Total sperm counted) × 100%.

### 2.7. Measurement of Sperm ROS Levels

Reactive oxygen species (ROS) levels in spermatozoa were quantified using a commercial assay kit (CA1420, Beyotime, Shanghai, China). Frozen-thawed semen samples were centrifuged at room temperature (500× *g*, 10 min) and the supernatant discarded. The sperm pellet was resuspended in phosphate-buffered saline (PBS) to a final concentration of 1 × 10^8^ cells/mL. A 200 μL aliquot of this suspension was incubated with 1 μL of 5 mM fluorescent dye stock solution at 37 °C for 30 min. Following incubation, samples were centrifuged (500× *g*, 10 min, RT) to remove residual dye, washed twice with PBS, and finally resuspended in 50 μL PBS for fluorescence measurement. Sperm smears were prepared onto glass slides, observed under fluorescence microscopy at 200× magnification, and digitally imaged. To ensure reliability and account for biological variability, the entire staining and evaluation process was performed with five independent replicates for each experimental group, using pooled sperm samples. Mean fluorescence intensity was quantified using ImageJ software (version 1.41o). For statistical analysis, the total fluorescence intensity of each image was normalized to the number of spermatozoa counted in the corresponding microscopic field, yielding a normalized fluorescence value per sperm cell that was used for all subsequent comparisons.

### 2.8. Metabolite Extraction

For metabolomic analysis, six independent biological replicates were processed per experimental group. Each biological replicate was constructed from the pooled semen sample created from ejaculates collected on different days. A separate straw from the same biological replicate was used. To preserve metabolite integrity, the straw was thawed slowly at 4 °C for 12 h, in contrast to the rapid-thawing protocol (38 °C, 30 s) used for sperm quality assessment. The entire content of one 0.25 mL straw (containing approximately 20 million sperm cells at a concentration of 8 × 10^7^ cells/mL) was added to 1 mL of a pre-cooled methanol:acetonitrile:water solution (2:2:1, *v*/*v*/*v*). After thorough vortex mixing, the mixture was subjected to ultrasonic disruption at 40 kHz for 20 min in a 4 °C water bath and then incubated at −20 °C for 1 h to precipitate proteins. Following centrifugation at 13,000 rpm and 4 °C for 15 min, the supernatant was collected and lyophilized for subsequent LC-MS/MS analysis, while the precipitate was reserved for proteomic analysis. All processed samples were stored at −80 °C until analysis.

### 2.9. Metabolome Sequencing and Data Analysis

The samples were analyzed using a Dionex Ultimate 3000 UHPLC system (Thermo Fisher Scientific, Waltham, MA, USA) coupled to a Q Exactive Puls high-resolution mass spectrometer (Thermo Fisher Scientific, Waltham, MA, USA). Chromatographic separation employed an ACQUITY UPLC HSS T3 column (100 × 2.1 mm, 1.8 μm) (176001131, Waters, Shanghai, China) at 45 °C, with mobile phases consisting of water (0.1% formic acid) and acetonitrile (0.1% formic acid) at 0.35 mL/min flow rate and 2 μL injection volume. Mass spectrometry conditions included electrospray ionization in positive/negative modes, scanning *m*/*z* 100–1000, with spray voltages of +3800 V/−3000 V, sheath gas 35 psi, auxiliary gas 8 psi, ion source temperature 350 °C, and ion transfer tube temperature 320 °C. Chengde hornless goat semen data were processed via Progenesis QI v2.0 software to extract and normalize retention times, *m*/*z* values, and peak areas. Multivariate statistical analyses included unsupervised PCA and supervised OPLS-DA. VIP > 1 and *p* < 0.05 criteria identified significant metabolites, with metabolic pathway analysis conducted using KEGG and MetaboAnalyst 4.0 to determine key biochemical pathways for differential metabolites. The metabolomics data have been deposited to MetaboLights (https://www.ebi.ac.uk/metabolights (accessed on 18 August 2025)) [23] repository with the study identifier MTBLS12860.

### 2.10. Protein Extraction

For each experimental group, three independent biological replicates were processed. Each replicate consisted of a pooled sample thawed at 4 °C, mixed with 1 mL liquid (methanol: acetonitrile: water = 2:2:1, *v*/*v*), and vortexed thoroughly. After low-temperature ultrasonication, samples were incubated at −20 °C for 1 h for protein precipitation, centrifuged at 13,000 rpm (4 °C, 15 min), and the pellets were sonicated at 40 kHz for 20 min to extract proteins.

### 2.11. Proteome Sequencing and Data Analysis

Following protein quantification, samples underwent tryptic digestion, TMT labeling, and SCX fractionation prior to LC-MS/MS analysis. Chromatographic separation used an EASY-nLC 1000 UHPLC system with mobile phase A (0.1% formic acid) and B (0.1% formic acid in acetonitrile). Gradient conditions were set as follows—0–40 min (6–20% B), 40–52 min (20–30% B), 52–56 min (30–80% B), and 56–60 min (80% B)—at a flow rate of 350 nL/min. Peptides were ionized via an NSI source (2.4 kV) and analyzed on an Orbitrap Fusion Lumos mass spectrometer. Full-scan MS spectra (MS1) were acquired across 350–1550 *m*/*z* at 60,000 resolution, while MS2 spectra started from 100 *m*/*z* at 15,000 resolution. In DDA mode, the top 20 most intense precursors were selected for HCD fragmentation using 32% NCE. Key parameters included an AGC target of 5e4, an intensity threshold of 5000 ions/s, a maximum injection time of 100 ms, and a dynamic exclusion duration of 30 s.

Database search and quantification: Raw files were processed using Proteome Discoverer™ 2.5 with MASCOT v2.8 against the UniProt *Capra hircus* database concatenated with common contaminants. Search parameters included trypsin digestion with a maximum of 2 missed cleavages, a precursor mass tolerance of ±10 ppm, and a fragment mass tolerance of 0.02 Da. Fixed modifications consisted of carbamidomethyl (C) and TMT 10plex (N-termini and Lysine), while variable modifications included oxidation (M) and TMT 10plex (Tyrosine). Protein quantification used unique peptides only, with normalization based on the median protein ratio across all samples. To ensure reliability and account for biological variability, the entire analysis was performed with three independent replicates for each experimental group, using pooled sperm samples.

Differential proteins between control and sericin groups were defined by ≥1.2-fold abundance change (*q* < 0.1). Bioinformatics analysis included GO and KEGG pathway annotation via DAVID 6.8, with cluster analysis and visualization performed in R (v3.4.4). Additionally, sperm proteome and metabolome sequencing were outsourced to Anoroad Gene Technology (Beijing, China) Co., Ltd. The mass spectrometry proteomics data have been deposited to the ProteomeXchange Consortium (https://proteomecentral.proteomexchange.org (accessed on 18 August 2025)) via the iProX partner repository [24,25] with the dataset identifier PXD067449.

### 2.12. qRT-PCR

Total RNA was extracted from sperm samples using Trizol method, with purity verified by NanoDrop 2000 (Thermo Fisher Scientific, Waltham, MA, USA). After genomic DNA removal, 1 μg RNA was reverse-transcribed using PrimeScript™ RT Reagent Kit (RR047A, Takara, Beijing, China). It is important to note that spermatozoa are transcriptionally silent cells; therefore, the mRNA levels measured here represent residual transcripts that may reflect cellular responses to cryopreservation stress and treatment, rather than de novo gene expression. Gene-specific primers (Table 2) were designed via Primer 3.0 against NCBI coding sequences, with specificity confirmed by melt curve analysis and agarose gel electrophoresis. Quantitative PCR was performed in 20 μL reactions containing 10 μL GoTaq^®^ qPCR Master Mix (A6001, Promega, Beijing, China), 200 nM of each primer, and 5 ng of cDNA template. Amplification was conducted on a QuantStudio 6 Pro system (Thermo Fisher Scientific, Waltham, MA, USA) with the following cycling parameters—95 °C for 2 min; 40 cycles of 95 °C for 15 s and 60 °C for 60 s; followed by melt curve analysis from 65 °C to 95 °C. β-actin (ACTB) and GAPDH served as reference genes. Relative expression was calculated using the 2^(−ΔΔCt)^ method with three biological replicates, each including three technical replicates. Amplification efficiencies of 95–105% were confirmed by standard curve validation.

### 2.13. Statistical Analysis

Statistical analyses were performed using SPSS 23.0 (IBM Corp., Armonk, NY, USA). Continuous data are presented as mean ± standard deviation. The normality of data distributions was confirmed by Shapiro–Wilk tests, and homogeneity of variances was assessed using Levene’s test. Where parametric assumptions were met, one-way ANOVA followed by Tukey ‘s post hoc test for multiple comparisons, or Student’s *t*-test for two-group comparisons, was applied. Pearson’s correlation coefficient was employed to evaluate linear relationships between variables. Statistical significance was defined as *p* < 0.05.

## 3. Results

### 3.1. Effect of Sericin on Post-Thaw Sperm Viability

As shown in Table 3, supplementation with sericin in the semen cryodiluent significantly enhanced post-thaw sperm motility in Chengde hornless goats. The 0.6% sericin group achieved the highest motility (65.25%), though not statistically different from the 0.4% group (*p* ≥ 0.05). Kinematic analysis demonstrated significantly higher VSL, VCL, VAP, and ALH parameters in the 0.6% sericin group compared to controls (*p* < 0.05), with several indices superior to other treatment groups.

### 3.2. Formatting of Mathematical Components

Supplementation with sericin in the semen cryodiluent significantly reduced sperm abnormality rates in cryopreserved Chengde hornless goat semen (*p* < 0.05; Figure 1). The 0.6% sericin group achieved the lowest abnormality rate (9.47%), though no significant difference was observed between 0.2% and 1.0% concentrations (*p* ≥ 0.05).

### 3.3. Effect of Sericin on Post-Thaw Sperm ROS Levels

Compared to the control group, supplementation with sericin significantly reduced post-thaw sperm ROS levels (*p* < 0.05; Figure 2; see Appendix A). The 0.6% sericin group demonstrated the lowest ROS levels, which were significantly lower than those in the 0.2%, 0.4%, and 1.0% sericin groups (*p* < 0.05). However, no significant differences were observed between the 0.8% sericin group and the 0.2%, 0.4%, 0.6%, or 1.0% sericin groups (*p* ≥ 0.05).

### 3.4. Untargeted Metabolomics-Based Analysis of the Effect of Sericin Proteins on the Cryopreservation of Semen from Chengde Hornless Goats

Figure 3a,b reveal significant separation between the frozen control and sericin groups through distinct clustering patterns at *p* < 0.01 (*n* = 6). Metabolites with VIP > 1, *p* < 0.05, and fold change (FC) > 2.5 or <0.4 were defined as significantly upregulated or downregulated metabolites, respectively. A total of 109 differential metabolites were screened, including 62 significantly upregulated and 47 significantly downregulated metabolites, as visualized in the volcano plot (Figure 3c). Through combined analysis of differential metabolites and KEGG pathway enrichment (Figure 3d,e), glutamine and its associated alanine, aspartate and glutamate metabolism pathway were identified as highly significant.

Furthermore, correlation analysis of differentially expressed proteins revealed a strong positive association with taurine (*r* > 0.8, *p* < 0.001), which was further supported by significant enrichment of the taurine and hypotaurine metabolism pathway in KEGG analysis (Figure 3e,f). ROC analysis of these two differential metabolites (glutamine and taurine) validated their biomarker potential, with area under the curve (AUC) exceeding 0.95 (Figure 3g,h).

### 3.5. TMT-Tagged Quantitative Proteomics-Based Analysis of the Effect of Sericin Proteins on the Cryopreservation of Semen from Chengde Hornless Goats

Through data transformation, fold change (FC) calculation, and quantitative *t*-test, we analyzed differentially expressed proteins between the cryopreservation control group and the sericin group. Screening based on FC > 1.2 or <0.8 and *q* < 0.1 identified 85 upregulated and 77 downregulated differential proteins, visualized in the volcano plot (Figure 4a). Among the differential proteins, we identified numerous redox-related proteins such as, LRP8, GSTM3, and SIRT2 (Figure 4b).

The PPI network constructed using the STRING database showed interactions among 24 proteins including PRDX5, CANX, and PRDX3 through pathways like redox balance and cellular structure maintenance (Figure 4c). Furthermore, GO enrichment analysis comparing the cryopreservation control and sericin group revealed that differential proteins were primarily enriched in sperm-egg recognition and peptidase activity for clearing damaged proteins within sperm (Figure 4d). The integration of proteomic and metabolomic data through correlation analysis revealed significant cross-talk between specific proteins and metabolites. Notably, a positive correlation was observed between the expression of GSTM3 and the abundance of the amino acid glutamine (Figure 4e).

### 3.6. Verification of Sericin’s Antioxidant Effects via Key Proteins in Chengde Hornless Black Goat Sperm

Based on proteomic analysis, three antioxidant-related proteins (LRP8, an apolipoprotein receptor; GSTM3, a ROS-detoxifying enzyme; SIRT2, a redox-regulating deacetylase) were selected for validation at the transcriptional level (Table 4). QRT-PCR was used to quantify residual mRNA levels of these targets in cryopreserved spermatozoa with or without supplementation of 0.6% sericin. Results demonstrated significantly elevated expression (*p* < 0.05) of all three mRNA transcripts in the 0.6% sericin-treated group (Figure 5). Although mRNA levels do not directly reflect protein abundance, the upregulation of these antioxidant-related transcripts is consistent with the proteomic profiles and supports the notion that 0.6% sericin enhances the antioxidant capacity of spermatozoa by strengthening cryoprotective defenses at the transcriptional level.

## 4. Discussion

Oxidative damage represents a primary mechanism of cryodamage in spermatozoa, driven by excessive reactive oxygen species (ROS) generation and associated oxidative stress during cryopreservation [26]. Seminal ROS equilibrium becomes disrupted during goat sperm freezing, compounded by the plasma membrane’s high polyunsaturated fatty acid (PUFA) content [4]. While PUFAs facilitate sperm motility under physiological conditions, they also increase vulnerability to ROS-induced lipid peroxidation [5]. Consequently, oxidative balance critically determines post-thaw sperm quality. Current research demonstrates that exogenous antioxidant supplementation improves cryopreservation outcomes [27]. Sericin, a natural protein with a unique molecular structure and bioactivity, has shown significant cryoprotective effects. Kumar et al. [28] reported that 0.25–0.5% sericin in buffalo semen extenders enhanced post-thaw viability, kinematic parameters, acrosomal/membrane integrity, and antioxidant enzyme activity. Similarly, Reddy et al. [21] observed improved motility and membrane protection in goats with 0.25–0.5% (*w*/*v*) sericin supplementation. Our functional assessments demonstrated that 0.6% (*w*/*v*) sericin supplementation significantly improved key sperm quality parameters, with total motility increasing from 38.5% to 52.3%, progressive motility from 22.1% to 35.7%, viability rates increasing from 45.72% to 55.38%, and morphological abnormality rates decreasing from 15.14% to 9.47% compared to the control group. These measurable improvements in sperm motility and quality provide the functional context for our subsequent molecular investigations.

The observed functional improvements, particularly in motility parameters, were accompanied by molecular changes in both metabolic and proteomic profiles. The significantly improved motility observed in our study may be supported by substantial alterations in energy metabolism. The maturation of spermatozoa and their transit through the female reproductive tract require substantial energy [29]. Metabolic alterations involving adenosine, which may contribute to ensuring adequate energy supply [30], and metabolites such as alanine and glutamine, which may provide ATP through metabolic pathways including the tricarboxylic acid cycle [31], could provide the necessary energetic foundation for the enhanced motility we observed.

Additionally, our metabolomic analysis revealed significant changes in taurine metabolism, which may contribute to improved sperm function through multiple mechanisms. Beyond supporting energy metabolism, the identified metabolites may also contribute to oxidative protection. Glutamine has been associated with antioxidant functions [32], while taurine may improve antioxidant capacity by increasing SOD and T-AOC activity in spermatozoa [33]. Notably, taurine and hypotaurine metabolism emerged as particularly important pathways, potentially maintaining redox homeostasis and supporting membrane stability—both critical for maintaining motility during cryopreservation [34]. Betaine may offer additional protection through its antioxidant properties [35,36,37]. KEGG enrichment analysis revealed that these differential metabolites were associated with ABC transporter pathways, suggesting their potential roles in supporting the functional improvements we observed in motility, viability, and structural integrity. It should be noted that the primary metabolomics analysis employed a combination of *p* < 0.05 and VIP > 1 for feature selection, a threshold common in exploratory metabolomic studies. Therefore, the majority of the metabolic findings should be interpreted as generating hypotheses for future validation. However, we also applied FDR correction to these results, and several key metabolites, most notably glutamine (*q* < 0.05), remained significantly altered. This provides a higher degree of confidence in the specific involvement of these metabolites in the cryoprotective mechanism.

Our proteomic investigations complemented these metabolic findings and provided additional mechanistic insights for the improved motility. Bioinformatics screening identified proteins associated with antioxidant response and energy metabolism, including LRP8, GSTM3, and SIRT2 [38,39,40]. The upregulation of these proteins in response to sericin supplementation was associated with our observed functional improvements in post-thaw sperm motility and quality. LRP8, a member of the low-density lipoprotein receptor family, may be involved in sperm maturation processes [41,42]. GSTM3 has been proposed as a cryotolerance biomarker [43] and may help prevent lipid peroxidation [44,45], potentially contributing to the maintained membrane integrity, reduced morphological abnormalities, and improved motility we observed. SIRT2 may contribute to cellular protection by mitigating oxidative stress and DNA damage [46,47], which could underlie the improved viability and motility rates measured in our study.

To further explore these proteomic changes at the transcriptional level, we quantified the corresponding residual mRNA transcripts via qRT-PCR. It is important to note that spermatozoa are transcriptionally silent; therefore, the measured mRNA levels represent pre-existing transcripts whose stability may be influenced by the cryopreservation stress and sericin treatment, rather than de novo gene expression. Despite this, the observed concordance between protein abundance and residual mRNA levels provides valuable insights into the potential post-transcriptional regulation mechanisms underlying the cryoprotective effects of sericin.

Based on our integrated analysis of functional parameters and molecular profiles, we propose that sericin enhances sperm cryotolerance through a multi-faceted mechanism that primarily supports energy metabolism and oxidative defense. The significant improvements in sperm motility (35.8% increase in total motility; 61.5% increase in progressive motility) likely result from enhanced energy provision through altered alanine, glutamine and adenosine metabolism, coupled with protection of sperm structure and function through taurine-mediated membrane stabilization and GSTM3, LRP8, SIRT2 to associated antioxidant activities. This coordinated response appears to maintain functional integrity during cryopreservation stress, preserving both structural and kinetic properties of spermatozoa.

Our findings provide substantive evidence supporting sericin’s cryoprotective effects through measurable improvements in sperm quality parameters and correlated molecular changes. While further validation of the precise mechanisms is warranted, the consistent patterns observed across functional, metabolic and proteomic datasets strongly suggest that sericin preserves sperm function through coordinated modulation of energy metabolism and oxidative stress pathways. Future studies should establish direct causal relationships between these molecular changes and functional outcomes, particularly those related to fertility potential.

## 5. Conclusions

This study demonstrates that supplementing cryodiluents with 0.6% sericin significantly improves post-thaw sperm quality in Chengde hornless goats, as evidenced by enhanced motility parameters, reduced oxidative stress, and decreased morphological abnormalities. Integrated multi-omics analyses revealed the concomitant upregulation of cryotolerance-associated proteins (LRP8, SIRT2, GSTM3) and modulation of metabolic pathways involving glutamine and taurine. Based on these findings, we propose a testable model wherein sericin mediates cryoprotection through coordinated regulation of antioxidant defense systems and energy metabolism pathways. However, this proposed mechanism requires further validation through targeted functional studies. The results support the potential utility of sericin as a cryoprotective additive for goat semen preservation, while future research should establish direct correlations between these molecular changes and functional fertility outcomes.

## Figures and Tables

**Figure 1 animals-15-02830-f001:**
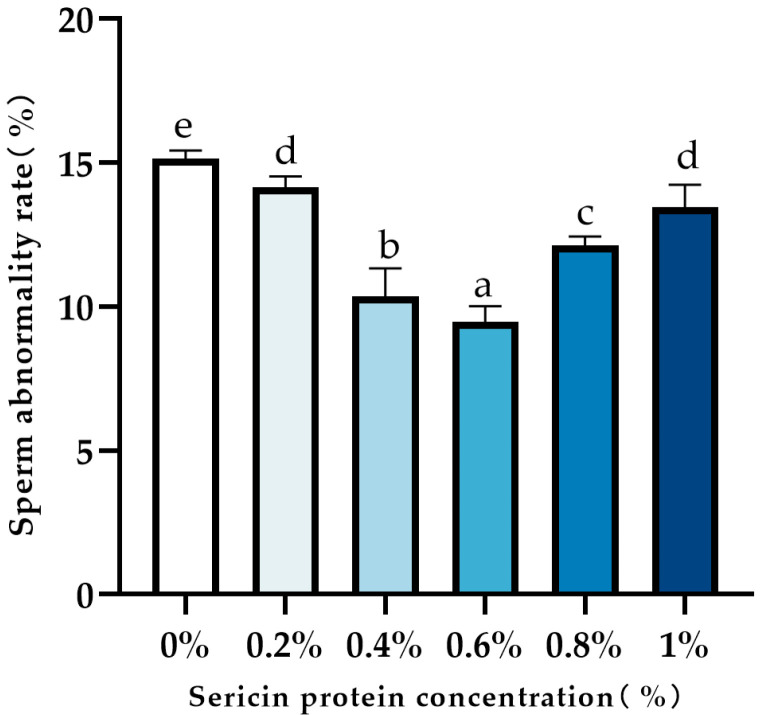
Effect of sericin on sperm deformity rate after freezing and thawing of semen from Chengde hornless goats. Data labeled with different lowercase letters denote significant differences (*p* < 0.05). Shared letters indicate non-significant differences (*p* ≥ 0.05).

**Figure 2 animals-15-02830-f002:**
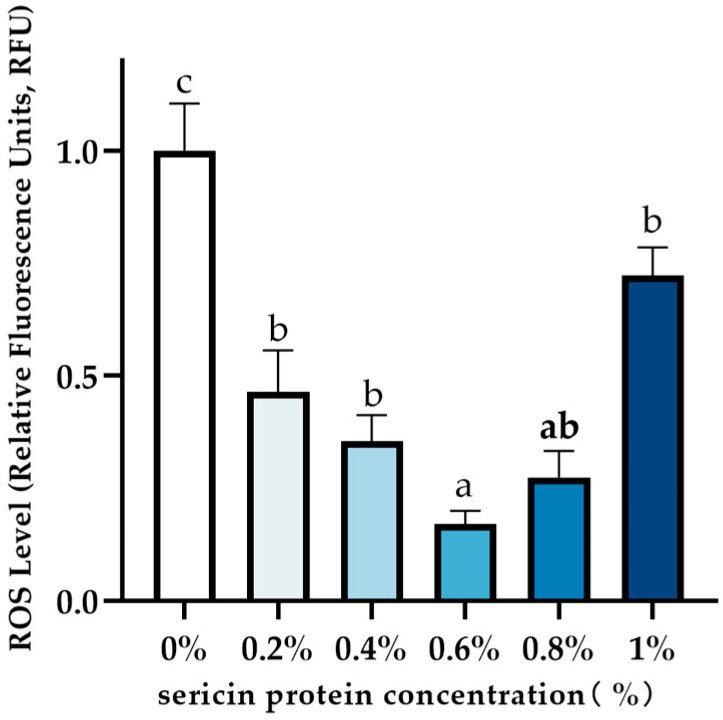
Effect of sericin on ROS levels after freezing and thawing of semen from Chengde hornless goats. Data labeled with different lowercase letters denote significant differences (*p* < 0.05). Shared letters indicate non-significant differences (*p* ≥ 0.05).

**Figure 3 animals-15-02830-f003:**
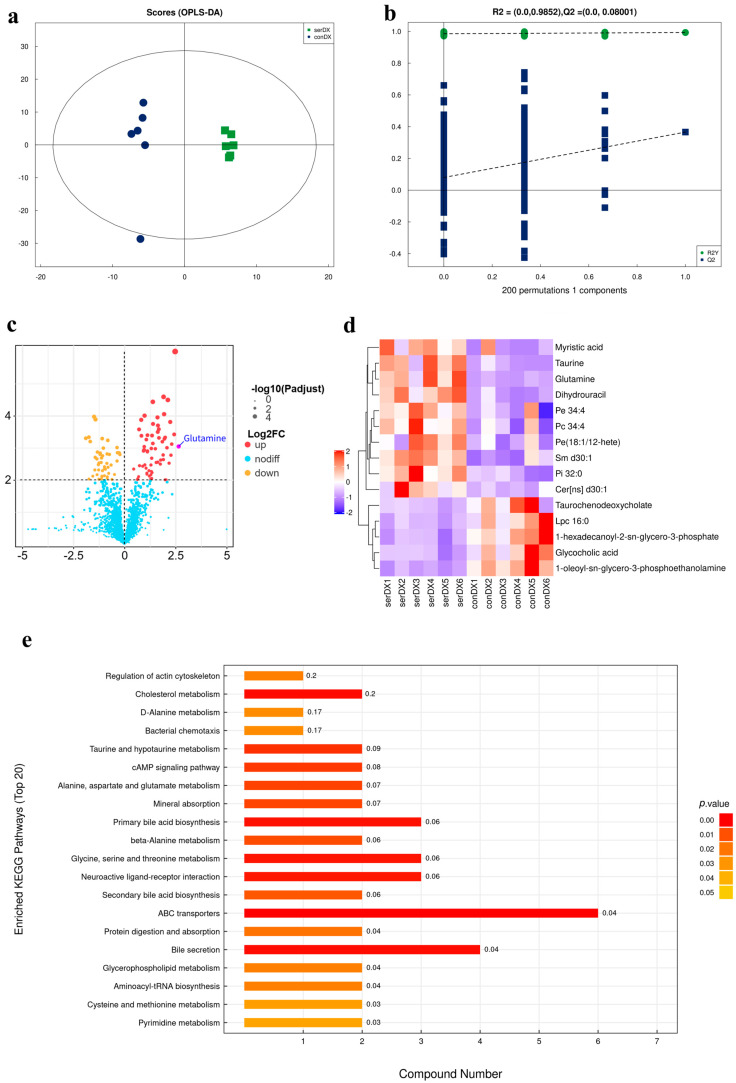
Metabolomic profiling of sericin-treated cryopreserved semen in Chengde hornless goats. (**a**) OPLS-DA score plots illustrating metabolic separation between groups. (**b**) Permutation validation plots for OPLS-DA models. The dashed lines represent the trend lines for the R2Y (green) and Q2 (blue) permutation data, respectively. (**c**) Volcano plot of differential metabolite screening. (**d**) Hierarchical cluster heatmap of significantly altered metabolites, conD1–conD6” denote six biological replicates of the untreated control group. “serD1–serD6” represent six biological replicates treated with 0.6% sericin. (**e**) KEGG pathway enrichment analysis of differential metabolites. (**f**) Correlation analysis between key metabolites. * represents significant difference, ** represents extremely significant difference, and *** represents more extremely significant difference. (**g**) ROC curve analysis of glutamine (**h**) ROC curve analysis of taurine.

**Figure 4 animals-15-02830-f004:**
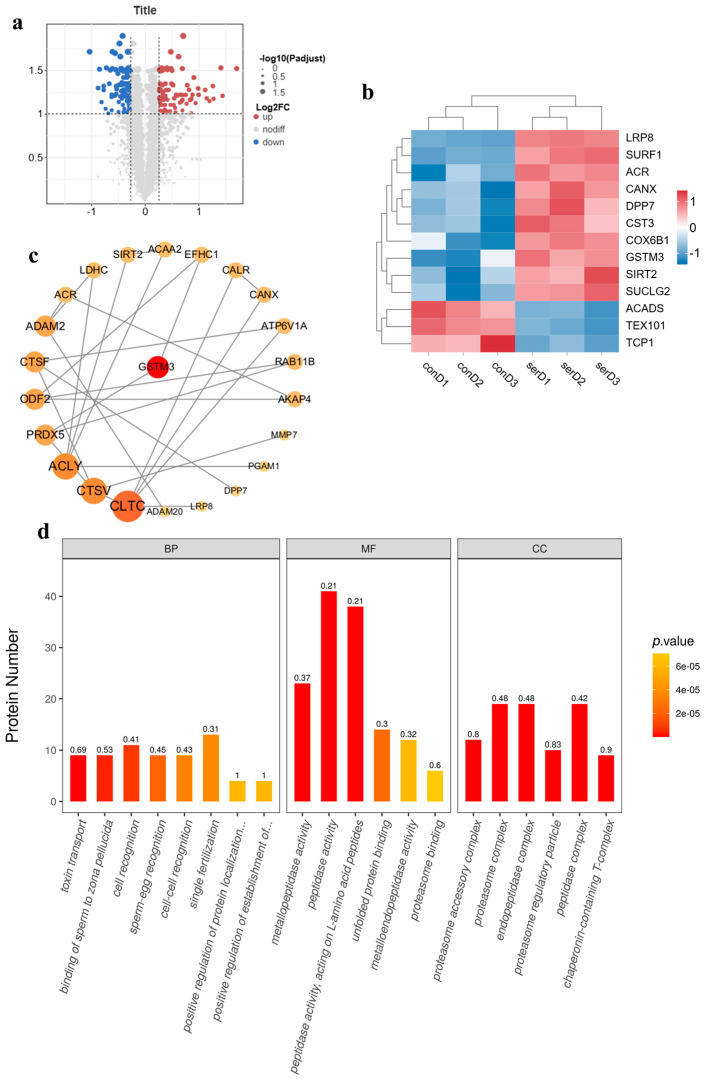
Proteomic analysis of sericin effects on cryopreserved Chengde hornless goat semen. (**a**) Volcano plot of differentially expressed proteins. (**b**) Cluster analysis of differentially expressed proteins, conD1–conD3” denote three biological replicates of the untreated control group. “serD1–serD3” represent three biological replicates treated with 0.6% sericin. (**c**) Protein–protein interaction network. (**d**) GO enrichment analysis of differentially expressed proteins. (**e**) Correlation heatmap integrating proteomic and metabolomic profiles. * represents significant difference, ** represents extremely significant difference, and *** represents more extremely significant difference.

**Figure 5 animals-15-02830-f005:**
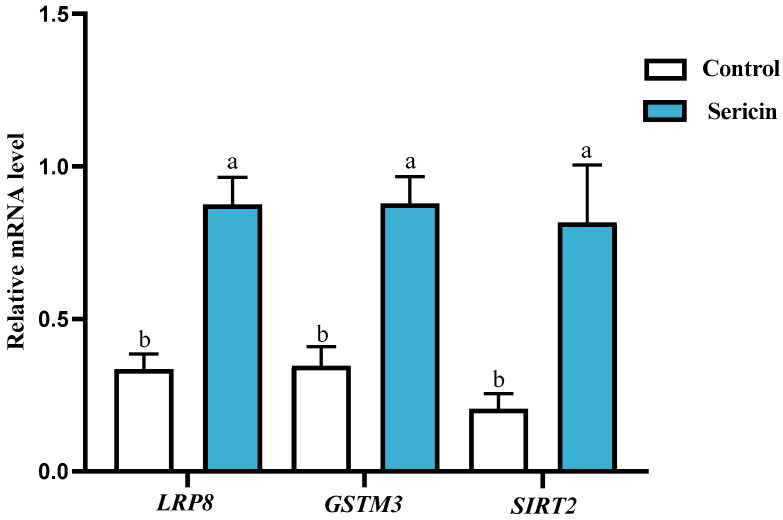
QRT-PCR analysis of mRNA expression. Data labeled with different lowercase letters denote significant differences (*p* < 0.05). Shared letters indicate non-significant differences (*p* ≥ 0.05).

**Table 1 animals-15-02830-t001:** The pH and osmolality of semen extender solutions supplemented with various concentrations of sericin.

Group	pH	Osmolality (mOsmol/kg)
Control	6.4	1645.67 ± 6.11
SER 0.2%	6.4	1652.67 ± 8.02
SER 0.4%	6.4	1645.33 ± 6.66
SER 0.6%	6.4	1652.33 ± 9.72
SER 0.8%	6.4	1658.33 ± 9.19
SER 1%	6.4	1659.67 ± 8.02

**Table 2 animals-15-02830-t002:** mRNA sequences of relevant genes.

Primers Name	Primer Sequences (5′-3′)
*LRP8*	F: CAAACGCCGATGTACCCT R: TGAGCCCGGACTTCTCAA
*GSTM3*	F: CCCAGAGCAATGCCATCTTG R: TGTTCCAAGTACCGAGGCTT
*SIRT2*	F: AAGGAGAAGACTGGCCAGAC R: GGAAGCTGAAGTAGTGGGGT
*β-actin*	F: CTCTTCCAGCCTTCCTTCCT R: GGGCAGTGATCTCTTTCTGC
*GAPDH*	F: ATGGCAAGTTCCACGGCACAGTC
	R: CAGCCTTCTCCATGGTAGTGAAG

**Table 3 animals-15-02830-t003:** Effects of sericin on sperm motility and kinetic parameters of Chengde hornless goats.

Movement Parameters	Control	SER 0.2%	SER 0.4%	SER 0.6%	SER 0.8%	SER 1%
TM/(%)	50.10 ± 1.69 ^d^	57.10 ± 1.79 ^c^	63.89 ± 2.43 ^ab^	65.25 ± 1.76 ^a^	60.24 ± 2.84 ^bc^	58.14 ± 1.25 ^bc^
VSL/(μm/s)	18.20 ± 0.50 ^d^	19.88 ± 0.51 ^c^	21.61 ± 0.71 ^bc^	22.90 ± 0.76 ^a^	20.81 ± 0.88 ^bc^	20.02 ± 0.23 ^c^
VCL/(μm/s)	49.54 ± 2.04 ^c^	53.55 ± 3.87 ^bc^	56.60 ± 2.66 ^ab^	59.73 ± 2.91 ^a^	54.27 ± 3.50 ^abc^	51.49 ± 2.43 ^bc^
VAP/(μm/s)	23.60 ± 1.64 ^c^	25.01 ± 1.76 ^c^	28.53 ± 3.13 ^bc^	33.17 ± 2.01 ^a^	29.33 ± 1.88 ^abc^	25.46 ± 1.43 ^c^
ALH/(μm)	11.98 ± 1.09 ^b^	13.22 ± 0.94 ^b^	13.75 ± 1.33 ^b^	16.30 ± 1.73 ^a^	14.12 ± 1.21 ^ab^	12.94 ± 0.75 ^b^

Different letters indicate significant differences between groups (*p* < 0.05).

**Table 4 animals-15-02830-t004:** Quantitative proteomic profiling of cryopreservation control and sericin-supplemented groups.

Protein Name	Control Group	Sericin Group	FC
LRP8	0.691	1133	1.641
GSTM3	0.869	1.094	1.259
SIRT2	0.903	1.112	1.231

## Data Availability

The original metabolomics data presented in the study are openly available in MetaboLights repository [23] with the dataset identifier MTBLS12860. The mass spectrometry proteomics data have been deposited to the ProteomeXchange Consortium (https://proteomecentral.proteomexchange.org) via the iProX partner repository [24,25] with the dataset identifier PXD067449.

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
