# Peer review of "Sericin Enhances Cryopreserved Sperm Quality in Chengde Hornless Black Goats by Increasing Glutamine Metabolism"

_animals, 2025, doi:10.3390/ani15192830_

Round 1
Reviewer 1 Report
Comments and Suggestions for Authors
Reviewer’s Comments on the Manuscript:
Title: Sericin Enhances Cryopreserved Sperm Quality in Chengde Hornless Black Goats by Increasing Glutamine Metabolism
The manuscript is interesting and timely, as it evaluates the role of Sericin supplementation on post-thaw semen quality in bucks and attempts to explore the underlying pathways. The work is scientifically relevant and involves the application of newer techniques, which adds value. However, certain aspects of the manuscript require substantial revision before it can be considered for publication.
General Comments
-
The study design is promising, but clarity in methodology and deeper integration of results with functional semen quality parameters are needed.
-
The abstract and discussion need major revision for coherence, balance, and scientific rigour.
-
Literature coverage can be broadened to highlight the novelty and importance of the study.
Specific Comments
1. Title
-
Appropriate and aligned with the content of the manuscript. No change suggested.
2. Summary
-
Needs modification as suggested and marked in the manuscript.
3. Abstract
-
Should be rewritten to provide:
-
A brief yet clear description of the experimental design.
-
Explicit presentation of results, including levels of statistical significance.
-
A clear conclusion specifying which concentration of Sericin was most effective.
-
4. Introduction
-
Literature review requires strengthening. Suggested inclusions:
-
Yadava et al. (2018): Effect of sericin supplementation on the semen quality of cryopreserved Hariana bull semen. Rumin. Sci. 7(1), 93–96.
-
Relevant studies by Reddy et al. and others on gene expression analysis in cryopreservation.
-
-
The knowledge gap should be explicitly mentioned.
-
Emphasise the importance of molecular studies in understanding Sericin supplementation effects.
5. Materials and Methods
-
Experimental design is not clearly presented. Authors should specify:
-
Number of samples collected per animal.
-
Number of freezings performed.
-
Season(s) during which semen was frozen.
-
Whether there were inter-buck differences in seminal attributes.
-
-
Using only three biological replicates appears insufficient for the robustness of the study.
6. Results
-
Figures 1a and 2a are not required; they may be omitted to avoid redundancy.
7. Discussion
-
Currently focused mainly on proteomic and metabolic pathways.
-
Functional parameters (e.g., motility, viability) are not adequately discussed in the context of results.
-
A major overhaul of the Discussion is required to balance molecular findings with functional semen quality outcomes.
8. Conclusion
-
Adequately written and aligned with the findings, but can be sharpened once Abstract and Discussion are revised.
Recommendation:
Major Revision
The manuscript has scientific merit but requires significant improvement in Abstract, Methods, Discussion, and integration of functional parameters before acceptance.

Reviewer 2 Report
Comments and Suggestions for Authors
This manuscript addresses the cryopreservation of Chengde polled goat semen, with a particular focus on the protective role of sericin supplementation. The authors demonstrate that adding 0.6% sericin improves post-thaw sperm viability, decreases abnormalities and ROS levels, and provides molecular insights through integrated proteomics and metabolomics analyses. Although sericin has been previously tested as a supplement in goat semen cryopreservation, this study, to my knowledge, is the first to explore in depth the mechanisms underlying its beneficial effects on post-thaw sperm quality.
The work is overall correct and relevant. However, several aspects need revision to improve scientific accuracy and clarity. In particular, the Results section frequently includes interpretative statements (e.g., possible roles of proteins or pathways, suggested mechanisms) that would be more appropriately placed in the Discussion. I also recommend increasing the size of all figures to facilitate visualization and revising figure legends to ensure that all abbreviations and experimental conditions are fully explained. Furthermore, throughout the manuscript, several bibliographic references are not preceded or followed by a space, which should be corrected for consistency. Finally, the qRT-PCR experiment targeting selected genes should be clearly presented as a validation step of the proteomic results, rather than in parallel with the omics analyses. Presenting metabolomics and proteomics first, followed by qRT-PCR validation, would improve the logical flow of the Results section.
Here are some specific suggestions that should be considered for future submissions:
Keywords
To promote the dissemination and scope of your work, it is recommended to include terms beyond those used in the title.
Introduction
Lines 51-53: Please consider using a period (.) instead of a colon (:) for better sentence structure. Are there any references to support your comment that this breed has low post-thaw semen quality and low fertility rates after artificial insemination?
Lines 58-60: What about the bibliographic reference?
Line 52: Same as in the previous comment.
Lines 67-69: According to which study?
Line 78: A period (.) is missing before “Here”.
Materials and methods
Line 84: If I understood correctly, the study was conducted on Chengde Hornless Black Goats, so the correct term here would be “buck” or “male goat.” The term “ram” is used to refer to males of the ovine species.
Lines 85-86: Which are the standardized breeding conditions and consistent semen collection protocols?
Lines 97-104: Normally, the minimum quality criteria that ejaculates must meet to be included in the pool should be discussed before mentioning the aliquots that are made with that pool. To which final concentration was the semen frozen?
Lines 127-128: Which magnification was used to observe the samples?
Lines 138-40: Resuspended to what concentration? At what magnification were the samples observed?
Lines 143-144: So, another straw is used for metabolite assessment, which is slowly thawed at 4 °C? What is meant by an adequate sample size? Is a specific number of sperm used?
Lines 145-148: Please rewrite this sentence to improve its understanding.
Line 150: Replace “analysed” with “analyzed”.
Line 195: Please note that, according to SI conventions, the correct abbreviation for seconds is “s”, not “sec”. I recommend replacing “sec” with “s” throughout the manuscript to ensure consistency with scientific standards.
Line 208: According to standard statistical reporting conventions, “p” values should be written with a lowercase italic “p” (e.g., p < 0.05), rather than with an uppercase or non-italic ‘P’. Please revise accordingly throughout the manuscript.
Results
Line 217: The title of Table 2 would be better as “Effects of sericin on sperm motility and kinetic parameters of Chengde hornless goats”. Please review and correct the units for the different sperm velocities within the table, as they must be expressed in μm/s.
Lines 228-229: The current graphics make it very difficult to see the colored arrows.
Lines 231-232: Although the conventions apply to the following figures, you should not omit this information, as all figures should be self-explanatory.
Lines 235-236: This statement is not entirely accurate, as the concentrations of 0.2% and 1% exhibited similar ROS levels to the control, as shown in Figure 2.
Line 239: The graph should include the ROS units.
Line 265: The figures should be separated or presented differently, as it is tough to visualize them in this way.
Lines 283-288: The statements referring to the potential role of GSTM3 and the suggested glutamine–GSTM3–glutathione axis constitute an interpretation rather than a factual description, which bibliographic references should also support. This content would be more appropriate in the Discussion section, while the Results should focus on objective findings.
Lines 294-298: Same as in the previous comment.
Line 299: In the Heatmap of Figure 4d, what do conD1, conD2, etc., mean? I assume they are the three replicates of the control group, but what concentration were the sericin-treated ones? All this should appear in the figure footnote.
Lines 315-327: Once again, all these lines are more consistent with a Discussion than with a presentation of Results and are not supported by any bibliography.
Discussion
Lines 335-339: And the references for all this?
Line 356: “To maintain sperm motility” sounds repetitive, as at the beginning of the sentence already mentions that sperm consume energy during motility and metabolism.
Lines 356-360: Rewrite to improve clarity and avoid repetition.
Lines 382-382: Which study states this?
Reviewer 3 Report
Comments and Suggestions for Authors
Recommendations for Authors
We appreciate the effort to combine phenotypic sperm analyses with proteomic and metabolomic approaches to explore the effect of sericin supplementation on goat semen cryopreservation. The topic is timely and of practical relevance for animal reproduction. However, several methodological issues, inconsistencies, and overinterpretations must be addressed before the manuscript can be considered for publication.
Below we provide detailed comments, organized by section.
Title, Simple Summary, and Abstract
- Please moderate causal statements such as “establishing a multidimensional framework” or “providing the molecular basis,” since the study is observational and lacks functional validation or fertility outcomes.
- Clarify the experimental unit (pooled ejaculates vs. individual males) in the Abstract. Current wording may overstate independence of replicates.
- Ensure consistency between numbers reported in Abstract (proteins, metabolites) and those in Results.
Introduction
- The rationale for using sericin is well described, but please also discuss physicochemical effects (viscosity, osmolality, pH) of adding sericin, which may influence CASA or ROS measurements independently of antioxidant action.
- Provide a more specific hypothesis (e.g., targeting glutamine/taurine metabolism) rather than a generic antioxidant expectation.
Materials and Methods
- Pooling: The semen was pooled across 6 bucks prior to treatments. After pooling, the experimental unit becomes the pool, not the individual male. Current analysis risks pseudoreplication. Please clarify and justify.
- Extenders: Report osmolality and pH of each extender after sericin supplementation.
- Cryopreservation protocol: Include cooling/freezing rates, number of straws per group, and criteria for selecting straws for analysis.
- CASA: Provide complete system settings (chamber type/depth, fps, temperature, sperm concentration).
- Morphology: Justify the use of Coomassie stain for morphological abnormalities, as this is not the most common method.
- ROS assay: Specify fluorophore and final concentration. Include positive control (e.g., H₂O₂) and autofluorescence controls with sericin. Indicate how fluorescence was normalized to cell number/field.
- Proteomics/Metabolomics:
- Provide database version, search engine, FDR thresholds (PSM, peptide, protein), normalization method, and details of TMT quantification.
- Apply FDR correction to differential expression (proteins and metabolites). Current FC+p-value thresholds are insufficient.
- qRT-PCR: Using ACTB alone in spermatozoa (transcriptionally silent cells) is problematic. Please justify or include additional reference genes/spike-in controls. Clarify that qPCR validates residual mRNA, not protein abundance.
- Statistics: Duncan’s test is not recommended; consider Tukey or Holm–Šídák. Adjust correlations for multiple testing.
Results
- Motility Table: Correct unit errors (μm/s, not μmol·(L·s)−1).
- Section Titles: Remove template artifacts (e.g., “Formatting of Mathematical Components”).
- Figures: Figure numbering is inconsistent — “Figure 1” appears twice. Please correct.
- Proteomics/Metabolomics: Report actual n per group, indicate whether results are based on pooled samples, and include full lists in Supplementary Data with statistics and FDR.
- Validation: The manuscript mentions validation by Western blot, but no methods/results are provided. Either remove these claims or include complete details (antibodies, blot images, quantification).
Discussion
- Avoid overstating causality. Present proposed pathways (glutamine–glutathione, taurine metabolism, LRP8/GSTM3/SIRT2 axis) as hypotheses.
- The role of LRP8 should be described more accurately (apolipoprotein receptor, not “oxidant-sensing receptor”).
- Temper conclusions about implications for artificial insemination since no fertility outcomes were assessed.
Conclusions
- Limit conclusions to directly demonstrated results (improved motility, lower ROS, fewer abnormalities with 0.6% sericin under the reported conditions).
- Reframe “multidimensional framework” as a model requiring further validation.
Data Availability and Transparency
- Please provide accession numbers for proteomic (PRIDE) and metabolomic (MetaboLights) datasets.
- Upload raw CASA data, ROS measurements, and complete metabolite/protein lists as Supplementary Materials.
Technical and Editorial Issues
- Correct all unit errors (μm/s).
- Fix figure numbering (avoid two “Figure 1”).
- Remove leftover template phrases (“Formatting of Mathematical Components”).
- Ensure consistent nomenclature of the breed (“Chengde hornless black goat”).
The manuscript is generally understandable, but the English requires improvement to ensure clarity and scientific precision. Several sentences are long, redundant, or awkwardly structured, which makes interpretation difficult. There are inconsistencies in terminology (e.g., breed names, protein functions) and unit reporting (e.g., μm/s incorrectly written as μmol·(L·s)−1). The use of technical terms should be standardized, and some overstatements need to be moderated. A thorough language revision by a proficient English speaker with expertise in scientific writing is strongly recommended.
Round 2
Reviewer 1 Report
Comments and Suggestions for Authors
Authors have clarified all the queries and revised the manuscript accordingly.
Reviewer 2 Report
Comments and Suggestions for Authors
The revised version of this manuscript shows considerable improvement compared to the initial submission, and I appreciate the effort the authors have made to address the previous comments. Nevertheless, further changes are still necessary before the manuscript can be considered for publication. Some formatting issues remain unresolved, such as missing spaces before and after bibliographic references, and scientific names that are not written in italics (e.g., Lines 72 and 249). In addition, several points still require clarification or correction, as detailed below.
Introduction
Line 84: Please specify whether the percentage of sericin reported in that study refers to v/v or w/v, as you do in your own study. This will allow the reader to interpret the comparison without consulting the cited article. This applies to the entire manuscript.
Materials and methods
Line 128: From the text and the osmolarity values in Table 1, it seems that the data correspond to the diluted semen samples in each experimental group, not just to the extender as stated in the title. Please modify this accordingly. Was any statistical analysis performed? Significant differences may exist between extreme groups, rather than only “minor variations.”
Line 136: Only 600 million spermatozoa per mL?
Line 164: In what medium were the samples diluted to reach this concentration? Also, please specify the magnification used to observe the samples.
Line 173: To what concentration were the samples resuspended? Moreover, the number of sperm initially taken for staining is not indicated, so it is not possible to deduce the actual number of sperm analyzed.
Results
Line 298: Some units are enclosed in brackets, while others are not. Please standardize the format.
Lines 308-309: And how do you interpret p = 0.05? Do you consider it statistically significant or not? This should be stated clearly and applied consistently throughout the manuscript.
Lines 311-320: The written description of the results does not match what is shown in the graph, which also differs from the one in the initial submission. Please review this carefully.
Figures 1 and 2: Some elements appear in bold, which looks inconsistent and visually unappealing. Please unify the formatting.
Lines 365-368: Again, Interpretative statements and assumptions should not be included in the Results section.
Reviewer 3 Report
Comments and Suggestions for Authors
The revised version of your manuscript represents a substantial improvement in terms of clarity, methodological rigor, and moderation of claims. Most of the issues raised in the first review round have been addressed satisfactorily. The causal language in the Abstract and Introduction has been moderated, the experimental unit is now clearly defined as pooled ejaculates, and the physicochemical effects of sericin supplementation (pH, osmolality, viscosity) are properly described with values reported. The Methods section has been considerably strengthened with detailed information about the cryopreservation protocol, CASA system configuration, staining choice, ROS assay including fluorophore, positive and autofluorescence controls, as well as proteomics and metabolomics pipelines, replicates, and statistical corrections. Figures and tables have been corrected regarding units, numbering, and formatting artifacts, and the qRT-PCR analysis has been repeated using GAPDH as a reference gene, which improved the robustness of the data and is now reflected in both methods and figures.
Despite these improvements, two minor issues remain. First, while the rationale for adopting FDR < 0.1 in proteomics is clearly explained, the metabolomics analysis is still based on p < 0.05 combined with VIP > 1. This discrepancy in statistical thresholds may raise concerns, and it would be helpful to explicitly acknowledge in the manuscript that metabolomics results are exploratory and should be interpreted with caution. Second, although your response correctly points out that spermatozoa are transcriptionally silent and that qRT-PCR therefore measures only residual mRNA, this limitation should also be clearly stated in the manuscript text itself, either in the Methods or the Discussion, to avoid any possible misinterpretation.
Overall, the manuscript has improved significantly and is now close to publication standard. Addressing these two points with minor revisions will further enhance transparency and strengthen the scientific soundness of your work.
